# COVID-19 Surveillance through Twitter using Self-Supervised and Few Shot Learning

**Brandon Lwowski**[1] **and Paul Rad**[2]
Department of Information Systems and Cyber Security
University of Texas at San Antonio
San Antonio, TX 78249, USA
[1]`oek437@my.utsa.edu` and [2]`Peyman.Najafirad@utsa.edu`

## Abstract

Public health surveillance and tracking virus via social media can be a useful digital tool for contact tracing and preventing the spread of the virus. Nowadays, large volumes of COVID-19 tweets can quickly be processed in real-time to offer information to researchers. Nonetheless, due to the absence of labeled data for COVID-19, the preliminary supervised classifier or semi-supervised self-labeled methods will not handle non-spherical data with adequate accuracy. With the seasonal influenza and novel Coronavirus having many similar symptoms, we propose using few shot learning to fine-tune a semi-supervised model built on unlabeled COVID-19 and previously labeled influenza dataset that can provide insights into COVID-19 that have not been investigated. The experimental results show the efficacy of the proposed model with an accuracy of 86%, identification of Covid-19 related discussion using recently collected tweets.

## 1 Introduction

The typical seasonal influenza virus and the current development of COVID-19 have multiple similarities from symptoms to how the virus is spread. Both viruses attack the respiratory system, can be spread through asymptomatic carriers, cases can range from mild to severe cases, and are transmitted by contact and/or droplets. Influenza and COVID-19 both can impact a community negatively due to the contagious nature of the virus and the high number of deaths caused by the viruses.

Public health surveillance like digital contact tracing (Ferretti et al., 2020; Ekong et al., 2020), epidemiological studies (Salathé et al., 2013), event detection (Lwowski et al., 2018), and monitoring the prevalence of vaccinations (Huang et al., 2017) can be used to help contain the virus and prevent its spread to the masses. These tools and techniques range from cellphone applications installed on personal phones that track the exact spread of a virus (Ekong et al., 2020) to the development of machine learning-based techniques to study the spread of a virus using social media (Lamb et al., 2013; Corley et al., 2009, 2010; Santillana et al., 2015; Broniatowski et al., 2013; Signorini et al., 2011). Similarly, machine learning-based methods have been developed to monitor the public's view on vaccines to combat the anti-vaccine narrative (Huang et al., 2017).

Using large sources of public information from social media to mine influenza and COVID-19 data allows researchers to help gain insight about the viruses. Just using a search word like "flu" and "coronavirus" with the Twitter API will return millions of tweets with information about vaccines, rumors, symptoms, and family/friends who have contracted the virus. Classifying tweets in to smaller subsets including categories like "Self vs Other" and "Awareness vs Infection" provides a deeper understanding on how the the influenza and COVID-19 are affecting the communities. Example tweets of each category can be found in Table 1

While other researchers aim to use unsupervised learning to cluster and perform topic modeling on COVID-19 tweets (Mackey et al., 2020; Medford et al., 2020), we decided to combine self-supervised learning combined with few shot learning to produce more accurate predictions for specific categories.

The major road block for using deep learning models on the COVID-19 tweets is the lack of annotated data. With millions of tweets related to COVID-19 flooding social media, researchers have a difficult time performing supervised learning on the data. We propose a method to attack this problem by transferring knowledge learned in influenza data and integrating it with latent variables obtained from the unlabeled dataset of COVID-19 to preform a deeper understanding through self-

| COVID-19 Tweets | | |
|---|---|---|
| **Category** | **Tweet** | **Importance** |
| **Related** | - Is US using this HIV antiretroviral drug to treat corona. Seems this is very successful in treating patients in Kerala, India and they have high success rate #coronavirus #CoronavirusUSA
- GREAT NEWS! Mom just woke up and asked for soup. She's eating soup and drinking water, dad is still on the vent fighting the fight #COVID19 | Be able to classify the tweets into Related tweets allows researchers to filter out tweets that mention COVID-19 in order to attract consumers (Click Bait). |
| **Awareness** | - U.S. Citizens that are NOT showing #COVID19 symptoms are PLEADING to be let off cruise ship that has #Coronavirus infected passengers on it. I Wonder if these same passengers spoke up when all those images of Mothers & Kids being seperated &
- #COVID19 is such a public health threat because the virus can be transmitted by individuals who are infected, but are not showing symptoms. | Tweets that are classified as Awareness is beneficial to researchers when not wanting to look at information regarding users actually becoming infected with the virus. These topics can include perception of masks, rumors of vaccines, etc. |
| **Infection** | - I'm absolutely broken! This morning I found out my bio mom (who lives in the UK) is infected; also has pneumonia. Her medical team has said to "prepare for worst case scenario." Well, here we are! She's going to die alone with her entire family in another country. F YOU #COVID19
- So my 80yo old dad tested positive. He is now on a ventilator. I need all your prayers that he pulls through this. #Covid19 #CoronaVirus | When using Social Media for public health surveillance and contract tracing. Having a classifier that can accurately classify tweets as a positive case can be extremely beneficial in discovering hotspots for the virus as well as other people they may have came in contact with |
| **Vaccine** | - @siggyflicker I understand that, but unless the clinical studies are throughly completed (hopefully sped up), we have to continue to be cautious - even then, unfortunately it's still not a vaccine that will prevent the contraction and spread of #COVID19

- I'm tired of hearing all the scary stuff about the virus. The news needs to include what strides we are making in treatments, a vaccine, or even a cure. #COVID-19 #coronavirus #CoronavirusUSA | Tweets aimed around the topic of Vaccines has present and future applications. In the present it allows researchers to investigate misinformation and/or the public perception of vaccines. In future research we can further classify these tweets into Intent to Receive or Already has Received a vaccine. |

Table 1: Example COVID-19 Tweets from the Coronavirus Tweets Dataset (Lamsal, 2020). Also provided in the table is a brief explanation of why the specific category is important to understanding multiple aspects of COVID-19. These categories allow researchers to study the virus at a more granular level.

supervised classification. The main contribution of this paper is three-fold:

- We propose a self-supervised learning algorithm to monitor COVID-19 Twitter using an autoencoder to learn the latent representations and then transfer the knowledge to COVID-19 Infection classifier by fine-tuning the Multi-Layer Perceptron (MLP) using few shot learning.

- We evaluate the utility of Twitter data for COVID-19 surveillance by training the 4 four binary models, in a computing environment provided by Jetstream (Stewart et al., 2015), to classify tweets into 4 different categories, related to COVID-19, COVID-19 Infection, COVID-19 Self/Others Infection, and COVID-19 Vaccine.

- Lastly but not least, we transfer a pre-trained influenza MLP classifier to fine-tune the accuracy of the self-supervised model.

## 2 Methodology

In this section, we explain the set-up of our study and motivate the core components of the proposed COVID-19 self-supervised learning with less labeled COVID-19 data. We begin with the introduction of our three datasets and data annotation strategy in Section 2.1 and follow by describing the core components of our self-supervised learning, as shown in Figure 1. We first learn how to generate the latent representation of unlabeled COVID-19 tweets using self-supervised Convolutional Autoencoder model in Section 2.2. Subsequently, design our COVID-19 supervised downstream task with a pre-trained Influnza classification, due to symptom similarities, and fine-tuning of the model using COVID-19 Few shot learning in Section 2.3 and 2.4. We finally present our results and evaluation metrics in Section 3 followed by discussion and conclusion in Section 4.

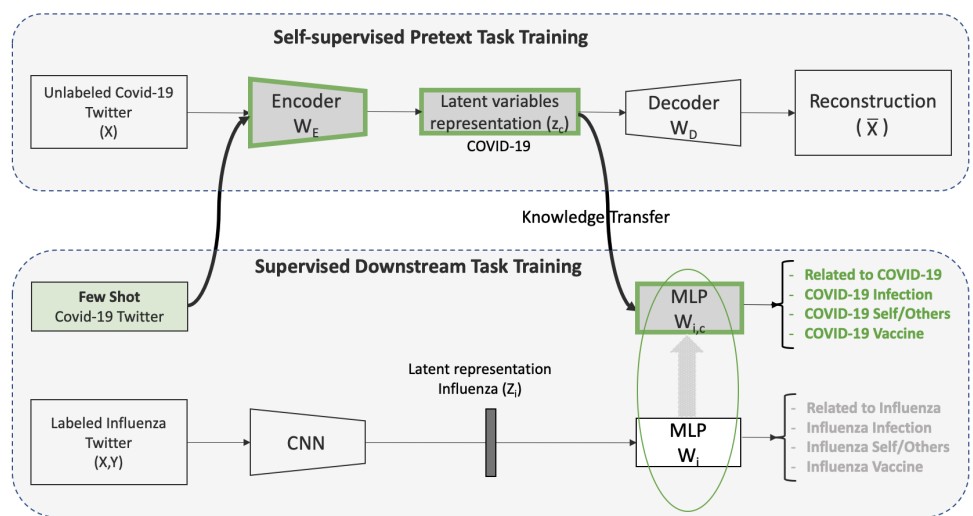

Figure 1: The proposed pipeline of self-supervised learning for COVID-19. The COVID-19 latent features are learned through the Autoencoder Convolution network to solve a pre-defined pretext objective. After self-supervised pretext training finished, the learned latent variables serve as a pre-trained model and are transferred to the downstream tasks by fine-tuning the Multi-Layer Perceptron(MLP) using few shot learning models after transferring knowledge learned from Influenza labeled dataset. The performance on the downstream objective is used to evaluate the quality of the learned features.

## 2.1 Datasets and Data Annotation

We make use of three datasets: Coronavirus (Lamsal, 2020) for self-supervised pre-task learning using Convolution Autoencoder, FluTrack (Lamb et al., 2013), and FluVacc (Huang et al., 2017) for training the influenza classifiers.

In order to use few-shot learning, an annotated COVID-19 dataset must be used to fine tune the overall model. The dataset collected by Lamsal et al (Lamsal, 2020) provides us with a large amount of data for analysis and predictions. a subsample of 500k tweets are used from this dataset to train the autoencoder. To obtain accurate annotations for this dataset, all tweets are shuffled and randomly sampled 25 times with a size of 100 tweets. Each sample is then distributed to 3 different annotators. Each annotator is asked to answer 4 questions about the tweet corresponding to the 4 categories we hope to classify. Only tweets where at least 2 annotators agree on the label are used for training and testing. A breakdown of each task and total number of tweets can be found in Section 3.4.

The previous work done by Lamb et al. and Huang et al. provided annotated datasets that allow us to implement supervised deep learning models. The FluTrack dataset provides 3 different classification labels: is the tweet related to the influenza or not, is the tweet talking about awareness or infection of the flu, and finally is the tweet about the user

or about someone else.The FluTrack dataset consisted of 11,990 tweets collected from years 2009 through 2012. The FluVacc dataset also has 3 classification labels: is the tweet about the vaccine or not, does the tweet contain intent to get the vaccine or not, and lastly does the tweet have information saying the user already received the vaccination or not. The FluVacc consisted of 10,000 total annotated tweets. The provided tweet labels are not mutually exclusive meaning a tweet can belong to multiple categories. In the example of "My mom caught the flu, hopefully i dont catch it..." can be classified as Infection and Other.

## 2.2 Convolutional Autoencoder

Most of the self-supervised techniques for latent text representation rely on Transformer architectures that predict the next token. In this study, we generate Coronavirus (Lam-sal, 2020) latent text representation by utilizing Convolutional Autoencoder. Autoencoders are a special category of deep neural networks that are deliberately programmed to make output as close as possible to input. Instead of training to predict y given input x, the network will be trained in an unsupervised approach to replicate its own input x. The autoencoder, in figure 1, is composed of 2 parts, the encoder and decoder. The job of the encoder is to compress the data into the latent space and the decoder takes the latent space

as its input and attempts to reconstruct the original input. To simplify the autoencoder, we define it as a composite function, Equation 1, with the encoder $E$ and decoder $D$ with a loss function defined to minimize the difference between the input, $X$, and the output, $\bar{X}$.

$$\min{(\bar{X} - X)} = (D(E(X)) - X) \qquad (1)$$

Once an autoencoder is trained for Coronavirus dataset, the latent variable $z$ can be used to extract important features for COVID-19 tweets. As shown in figure 1, both the encoder and decoder are convolutional neural networks. We pass the vectorized tweet through a word embedding, the word embedding layer converts every word of the tweet into an n-dimensional vector. This converts the input dimensions into 2-D that can be fed to the Convolutional Autoencoders for our task.

Hadifar et al. (Hadifar et al., 2019) also uses an Autoencoder in order to help classify text. They also use the Autoencoder to pre-train the encoder, but instead of classifying text, they use KNN for clustering similar text with the learned latent space. Their claims support the use of Autoencoders to achieve a deeper understanding of short texts such as tweets.

The word embedding used in our research was the GloVe 50d trained on 6 billion words from Wikipedia(Pennington et al., 2014) with a max tweet length of being 50 words. The max length of the tweet was decided by calculating 2 standard deviation greater than the mean allowing more than 95% of the tweets to not be altered in size. And tweets that are less than 50 words are post padded with 0's, representing no word being present.

The next step in the encoder consists of three different 1 dimensional convolutions with kernel sizes of 2,3 and 4. These three operations are grouped together in Table 2. This allows the encoder to learn 2,3 and 4 word relationships which is important in encoding meaning and semantics into the latent space. The outputs from each of the 3 convolutional layers are passed through an activation function (ReLU) and concatenated together. One more convolutional layer is used as well as a ReLU layer before flattening the output. The output of the flattened layer is the latent space which is half the size of the original input. The encoder ends at the Flatten layer in Table 2 and thus start the process of the decoder.

The output (latent space $z_c$) of the encoder is

**Auto Encoder**

| Layer (type) | Output Shape | Param # |
|---|---|---|
| Input Layer | (None, 50) | 0 |
| Embedding Layet | (None, 50, 50) | 750050 |
| Conv1D (bigrams) | (None, 50, 16) | 1616 |
| Conv1D (trigrams) | (None, 50, 16) | 2416 |
| Conv1D (4-grams | (None, 50, 16) | 3216 |
| Concatenate | (None, 50, 48) | 0 |
| Conv1D | (None, 50, 25) | 3625 |
| Flatten | (None, 1250) | 0 |
| Reshape | (None, 50, 25) | 0 |
| Conv1D (bigrams) | (None, 50, 16) | 816 |
| Conv1D (trigrams) | (None, 50, 16) | 1216 |
| Conv1D (4-grams | (None, 50, 16) | 1616 |
| Concatenate | (None, 50, 48) | 0 |
| Conv1D | (None, 50, 32) | 4640 |
| Conv1D | (None, 50, 15000) | 1455000 |
| Softmax | | |

Table 2: The architecture, output shapes, and number of trainable parameters for the Auto Encoder.

then used as the input to the decoder. Similar operations are performed in order to reverse the operations and reconstruct the original vector/tweet. If the decoder can accurately reconstruct the original input then the latent space has learned and encoded the right information in a compressed format.

The decoder begins with the latent space being reshaped into a 2 dimensional vector. A convolution of 2,3,4 are performed on the latent space. The outputs are passed through the ReLU activation and concatenated in the same manner as the encoder. Where the decoder differs from the encoder is the final layer. The output of the final convolutional layer is passed through the softmax activation function. By minimizing the difference between the input vector and reconstructed vector we achieve our goal of generating Tweet latent representation. Once the Autoencoder is trained on Cornonavirus data,the decoder can be removed from the network, leaving the encoder and latent space to be used as its own model.

## 2.3 Influenza Classification

A similar CNN architecture found in Kim et al. CNN for text classification is used to train a influenza tweet classifier (Kim, 2014). The influenza

tweets are transformed to vectors using the same word embeddings in the auto encoder. During training a latent representation is learned, $Z_i$, from the convolutions of size 3,4 and 5 words. The latent representation is then passed to the Multi-Layer Perceptron, MLP $W_i$, and is trained with supervision to predict the correct label $Y$. In Figure 1, this can be seen on the lower section of the image. A description of each label is discussed in detail in Section 2.1. The accuracy of the influenza classifier is discussed in Section 3.4 as well as Table 3.

## 2.4 Few-Shot Classification

A large influence for our research comes from Bowman et al. (Bowman et al., 2015). Bowman et al. takes the parameters of a model that was trained on one dataset and trains a new model using a portion of the new dataset. Bowman et al. suggests that by introducing a corpus that is high quality, it can be used to transfer knowledge and learn sentence meanings that can improve downstream text classification tasks.

At this point we have 2 trained models, the autoencoder for COVID-19 tweets and the influenza tweet classifier for the 6 categories. We want to use as much knowledge learned in the COVID-19 latent representation and Influenza MLP layer to accurately predict on COVID-19 classification tasks, with limited COVID-19 labeled data. We accomplish this task with few-shot learning with a warm start. This is a similar methodology to Dirkson et al. (Dirkson and Verberne, 2019). They use the ULMfit to transfer knowledge between health and twitter data (Howard and Ruder, 2018). Where we differ from the ULMfit learning is we completely freeze the encoders parameters and only allow the MLP to be trained.

The same process used for the influenza classifier is used for the COVID-19 classifier. The encoder $W_e$ and latent variable $z$ are frozen and the decoder is removed. MLP$_{i,c}$ is appended to $z$ and used to classify COVID-19 tweets. If this model were trained from a cold start, the weights of MLP$_{i,c}$ would be randomly assigned during the first epoch of training and adjusted from there. This prevents any classification knowledge being transferred from the influenza to the COVID-19 tweets. Instead, the weights of MLP$_i$ are used to initialize the weights of MLP$_{i,c}$. This warm start training allows the model to use knowledge from the the influenza training. We can then train the model with

the few labeled COVID-19 tweets and fine tune the model for the downstream task of classifying all COVID-19 tweets.

## 3 Results

In table 3, the accuracy of the influenza, COVID-19 trained on a cold start, and COVID-19 trained with the weights initialized with the influenza MLP (COVID*) are given for each of the 4 tasks. The task of Received and Intent are removed from the COVID-19 classifiers since there are no vaccines currently available for the virus.

| Accuracy | | | |
|---|---|---|---|
| **Categories** | **Influenza** | **COVID** | **COVID\*** |
| Related | .82/.76 | .96/.83 | .91/.86 |
| Aware. vs Infect. | .85/.70 | .97/.70 | .98/.73 |
| Self vs Other | .79/.70 | .98/.81 | .96/.86 |
| Vaccine | .98/.97 | .96/.80 | .97/.72 |
| Intent | .94/.82 | N/A | N/A |
| Received | .91/.80 | N/A | N/A |

Table 3: Results for Influenza and COVID-19 Classifiers on the test set. Accuracies reported are in the format of (Train Acc./Test Acc. COVID column is the results from training the MLP from scratch (cold start), while COVID* is the results from training the MLP by initializing the MLP with the influenza classifier weights.

| Precision, Recall, F1 | | |
|---|---|---|
| **Categories** | **COVID** | **COVID\*** |
| Related | .71/.70/.71 | .69/.77/.73 |
| Aware. vs Infect. | .69/.68/.68 | .73/.72/.73 |
| Self vs Other | .81/.76/.77 | .86/.87/.86 |
| Vaccine | .77/.73/74 | .67/.65/.66 |

Table 4: Macro Precision, Recall, and F1 for COVID-19 Classifiers. Statistics are reported in the format of P/R/F1. COVID column is the results from training the MLP from scratch (cold start), while COVID* is the results from training the MLP by initializing the MLP with the influenza classifier weights.

### 3.1 Related

For classifying tweets into COVID-19 Related and COVID-19 Non-Related, we took a sub sample COVID-19 Tweets and a sub sample of influenza related tweets and combined them into one dataset. We trained the classifier on 1000 influenza and 400 COVID-19 tweets but, tested on 4000 influenza and 1400 COVID-19 tweets. A training accuracy

of 91% and testing accuracy of 86% were achieved as well as a precision score of 0.69, a recall score of 0.77 and a F1 score of 0.73. The same train and testing are done on the cold start model but achieved a higher train but lower test accuracy with a precision score 0.71, recall score 0.70 and a F1 score of 0.71. The COVID-19 MLP initialized with the influenza MLP outperforms the the cold start on 3 out of the 5 statistics including a 3% increase on test accuracy.

### 3.2 Awareness Vs Infection

Awareness vs Infection can be seen as a binary classification problem. We classified each COVID-19 into 2 more subcategories, is the twitter user aware of the virus or are they talking about themselves or others being infected. We trained the classifier with 250 awareness and 250 infection tweets and tested on 450 tweets in each category. An accuracy of 98% on the train set and 73% on the test set were achieved as well as a precision score 0.73, a recall score of 0.72 and a F1 score 0.73. The same training and testing were carried our on the cold start model but achieved a lower training and lower testing accuracy with a precision score 0.69, recall score 0.68 and an F1 score of 0.68. The COVID-19 MLP initialized with the influenza MLP outperforms the the cold start on all accuracy statistics including a 3% increase on test accuracy.

### 3.3 Self Vs Other

In self vs other, we try and classify the infection tweets into further categories. We would like to clarify whether the post is about the author, or if the author is concerned about the COVID-19 infection of others. We trained the classifier on 55 tweets labeled "Other" and 20 tweets labeled as "Self". The test set consisted of 45 tweets labeled "Other" and 30 tweets labeled "Self". An accuracy of 98% on the train set and 86% on the test set were achieved as well as a precision score of 0.86, a recall score of 0.87, and a F1 score of 0.86. The same training and testing were carried our on the cold start model but achieved a lower training and lower testing accuracy with a precision score 0.81, recall score 0.76 and a F1 score of 0.77. The COVID-19 MLP initialized with the influenza MLP outperforms the the cold start on 4 out of the 5 statistics including a 5% increase on test accuracy.

### 3.4 Vaccine

The last category was Vaccine related tweets. We would like to clarify whether the post is about vaccine or cure, or whether it is about certain facets of the virus. The train set had 65 non vaccine related tweets and 35 vaccine related tweets. The test set had 60 non vaccine and 35 vaccine tweets. An accuracy of 97% for train and 72% on the test as well as a precision score of 0.67, a recall score of 0.65, and a F1 score of .66. The same train and testing are done on the cold start model but achieved a lower train but higher test accuracy with a precision score 0.77, recall score 0.73 and a F1 score of 0.74. The COVID-19 MLP initialized with the influenza MLP was outperformed by the cold start on 4 out of the 5 statistics including a 8% decrease on test accuracy.

## 4 Discussion and Conclusion

Diving further into the data and investigating how the classifiers could be improved, the first step would be to improve the quality of the tweets labels. Several tweets in the related category had mislabeled gold labels. For example, *"@berniesanders thank you so very much @berniesanders for giving us hope as a nation for and end to this joke of a presidency thank you for all the fund raising and support that you have done for our country in the age of covid19 and good sir"* was given the gold label of not related to covid when the model predicted it was. The overall theme was presidential candidate Bernie Sanders, but it was still related to COVID-19. Another example, *"tell congress to put people first, demand paid sick leave for our most vulnerable workers covid-19"*. This tweet should be labeled awareness but was given the gold label of infected. This tweet should be labeled as "Awareness" but was given the gold label of "Infected". With more time to weed out accurate labels and provide higher quality data to the classifier would increase overall accuracy across the board. In future research we aim to use services like Amazon Mechanical Turk to label more data for us.

Our self-supervised methodology for classifying COVID-19 tweets with fewer labeled data has been developed to overcome the challenges of labeling massive COVID-19 Tweet data. At the time of this research, labeled COVID-19 datasets for supervised learning were not readily available. That being said, to achieve maximal results, providing the

deep learning models with large amounts of high quality annotations for learning would be ideal. Nevertheless, with unlabeled data available and small amount of annotated tweets, our research demonstrates that we can transfer knowledge from unsupervised latent representation and high quality datasets to similar domain classifiers using self-supervision and few shot learning.

Lastly, our original hypothesis of influenza tweets and COVID-19 tweets being extremely close in context was not entirely true. While COVID-19 and Influenza tweets have a lot of similarities there also exists differences in the themes of the tweets. Twitter users tend to flood their timelines with posts and re-posts of the politics, rumors, misinformation and news related to COVID-19 rather than symptoms and infections.

Looking at the results for each category, leveraging the COVID-19 self-supervised pretext training to produce COVID-19 latent representations from unlabeled data for supervise downstream COVID-19 classifier shows promising results. Although the test accuracy of classifier can still be improved, the warm start of COVID-19 outperforms the cold start model on 3 out of 4 tasks in terms of the expected accuracy , precision , recall and f1. With this experiment we show that using high quality annotated data in a similar domain, can be used with self-supervision and few shot learning to train classifiers on data where labels are limited. Rather than starting from scratch we can initialize the weights on the MLP with the knowledge learned during supervised training. We believe that the architecture and methodology provided in this research show that using self supervision and few-shot learning can overcome some of the challenges of data with limited annotations. Using the proposed model to label tweets can assist future researchers to investigate COVID-19 tweets at a more granular level.

## Acknowledgements

This work is partly supported by a grant from the Intelligence Community Centers for Academic Excellence (IC CAE) Program and the Open Cloud Institute at University of Texas at San Antonio (UTSA). The authors gratefully acknowledge the use of the services of Jetstream cloud.

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
