# OpenReview forum: "COVID-19 Surveillance through Twitter using Self-Supervised and Few Shot Learning"
_EMNLP/2020/Workshop/NLP-COVID — NLP-COVID19-EMNLP Oral_

### Official Review · AnonReviewer1 · 2020-09-10
**Well written paper with good explanations, proper motivation, clear objectives and results.**

**Rating:** 7
**Confidence:** 4

**Review:**

This paper proposes a learning algorithm which is semi supervised and can monitor twitter during COVID19. They also evaluate the use of twitter data for surveillance and improve accuracy of their model using transfer learning. The paper shows that high accuracy may be obtained from limited labelled data using few shot learning and can potentially overcome the need to have large datasets which are labelled.

Pros:

1. Very well written paper with proper descriptions and examples where needed.

2. Figures and tables are very self explanatory in nature with good notes o them.

3. Processes of dataset building and data annotation has been clearly explained in detail.

4. The process of classifying data into its labels such as awareness vs infection or self vs other and related vs non related has been explained well and has been analyzed with proper statistical metrics and tests.

5. The work is very relevant to the current situation and on going research


Comments


1. A minor comment, but may be a bit more light on to why the specific number of data-points were chosen for classification would be better.

2. Although a very well written paper, there are few mistakes in writing and grammar. Such as "The accuracy of the influenza classifier is DISCUSS in section 3.4" in page 5 and a few other minute errors.

---

### Official Review · AnonReviewer2 · 2020-09-12
**A well written paper with an interesting approach but requires several minor clarifications**

**Rating:** 8
**Confidence:** 4

**Review:**

The paper proposes an approach to overcoming the lack of annotated data from Twitter related to covid-19. The authors take advantage of the similarities between covid-19 and common flu and employ transfer learning to detect tweets related to covid-19 and classify them into several categories. The technical part of the paper is very well written and the intuition behind various transformations is explained in a clear and simple way. I also appreciated the comments on the extension and application of the approach for future monitoring once a vaccine becomes available.

Areas for improvement:
- The major point of confusion for me is whether the authors trained one model to perform multi-class classification (as claimed in line 141) or four binary models to tackle each problem separately (as it appears in sections 3.1-3.4). On a related note, it is unclear if the classes considered in the paper are mutually exclusive or if each tweet has multiple labels? A figure/table clarifying this for both coronavirus data and influenza data would be very helpful.

Minor comments:
- Lines 237-238: it is unclear what are the 5 categories? Both the introduction section (line 141) and figure 1 mention only 4 categories. I suggest specifying the categories here one more time for clarity.
- Line 239: please provide the size of the final annotated dataset used for few-shot learning.
- Just a suggestion to reiterate in line 339 that the latent representation is denoted by z.
- Section 2.3: the authors described the process of training 2 encoders, one on Coronavirus data and one on Influenza data. It is unclear which encoder the authors are referring to in line 391. Moreover, figure 1 creates the impression that W_i denotes the weights of the pre-trained MLP model. To avoid confusion I suggest the authors review the paper and make sure variables are properly introduced in the main text.
- Please confirm that the reported in Table 4 precision, recall, and f1 score values are calculated using the test set.
- There are a few typos in the text, for example in lines 272, 343, 433, 573. While it doesn’t affect the readability it would be nice to correct those.

---

### Official Review · AnonReviewer3 · 2020-09-17
**A paper on semi-supervised learning needs more validation**

**Rating:** 7
**Confidence:** 5

**Review:**

This paper used few shot learning to build a semi-supervised model on unlabeled COVID-19 tweets. A pre-trained Influenza model was used to facilitate the classification tweets into related to COVID, Self/Others infection, Infection and Vaccine.  The paper is well motivated and clearly written.

Some limitations of the papers:

The test datasets used in the categories of Self/Others infection, Infection and Vaccine were very small. The benefit of semi-supervised learning is to use a small amount of labeled data to classify a large amount of unlabeled data. The comparison between COVID and COVID* using the pre-trained Influenza model in Table 3 and 4 can not be justified based on such small test datasets.

There is no baseline method in the paper. Many other semi-supervised methods such as Graph Convolutional Network (GCN) achieved good performance on small training sets. The purposed method needs to be compared with baselines including some traditional machine learning methods.

Some information needs to be clarified:

Figure 1. whether CNN used in Influenza classification is same as the autoencoder used for unlabeled COVID tweets. Please refer to Section 2.3 “Once a second Autoencoder on influenza data has been trained”.
Please provide the number of unlabeled COVID-19 tweets trained in autoencoder and labeled Influenza tweets used in classification.